# Differential Back Muscle Flexion–Relaxation Phenomenon in Constrained versus Unconstrained Leg Postures

**DOI:** 10.3390/bioengineering11070736

**Published:** 2024-07-20

**Authors:** Yi-Lang Chen, Ying-Hua Liao

**Affiliations:** 1Department of Industrial Engineering and Management, Ming Chi University of Technology, New Taipei 243303, Taiwan; m07218007@mail2.mcut.edu.tw; 2Taiwan Research Institute, New Taipei 251401, Taiwan

**Keywords:** trunk flexion, flexion–relaxation phenomenon, leg posture, flexibility, muscle activity

## Abstract

Previous studies examining the flexion–relaxation phenomenon (FRP) in back muscles through trunk forward flexion tests have yielded inconsistent findings, primarily due to variations in leg posture control. This study aimed to explore the influence of leg posture control and individual flexibility on FRP in back and low limb muscles. Thirty-two male participants, evenly distributed into high- and low-flexibility groups, were recruited. Activities of the erector spinae, biceps femoris, and gastrocnemius muscles, alongside the lumbosacral angle (LSA), were recorded as participants executed trunk flexion from 0° to 90° in 15° increments, enabling an analysis of FRP and its correlation with the investigated variables. The findings highlighted significant effects of all examined factors on the measured responses. At a trunk flexion angle of 60°, the influence of leg posture and flexibility on erector spinae activities was particularly pronounced. Participants with limited flexibility exhibited the most prominent FRP under constrained leg posture, while those with greater flexibility and unconstrained leg posture displayed the least FRP, indicated by their relatively larger LSAs. Under constrained leg posture conditions, participants experienced an approximate 1/3 to 1/2 increase in gastrocnemius activity throughout trunk flexion from 30° to 90°, while biceps femoris activity remained relatively constant. Using an inappropriate leg posture during back muscle FRP assessments can overestimate FRP. These findings offer guidance for designing future FRP research protocols.

## 1. Introduction

Musculoskeletal injuries often arise from repetitive tasks, improper force application, and awkward postures [1]. Notably, physical injuries can be linked to factors such as frequent exposure of the upper body to deep trunk flexion postures or prolonged stooping [2]. While there has been advocacy for squatting over stooping in the past, on-site field surveys reveal that workers commonly favor trunk flexion for its perceived efficiency [3,4,5,6].

Lower back pain (LBP) associated with trunk posture has been attributed to the flexion–relaxation phenomenon (FRP) in back muscles occurring during forward trunk bending [7]. FRP manifests when lumbar spinal muscles are supplanted by passive tissues of the spine, such as posterior spinal ligaments, to counterbalance trunk torque on the lumbar spine [8]. Excessive stretching of these passive tissues during FRP may result in increased lower back loads, as these tissues possess viscoelastic properties and are susceptible to creep deformation under sustained load [9,10,11], potentially resulting in cross-link rupture [12]. Zwambag and Brown [13] emphasized the significant role of passive tissues over back muscles in FRP, underlining its importance in understanding LBP mechanisms. To quantitatively evaluate the extent of FRP, the flexion–relaxation ratio was introduced, which measures the decrease in back muscle activity over a specific range of trunk flexion [14]. This approach allows for the comparison of FRPs across various controlled conditions.

Comprehending back muscle FRP has become pivotal for grasping LBP. Previous FRP studies have primarily focused on trunk forward flexion angle [15,16], individual flexibility [4,17], pelvic movement [18,19], and the presence of LBP [20,21]. During FRP experiments, participants typically execute forward trunk bending. However, control of the lower limbs and pelvic posture can also influence FRP during trunk flexion. Gupta [22] explored the relationship between FRP and lumbopelvic motion, discovering that restricting pelvic rotation led to early FRP occurrence due to reduced spinal stability and increased passive tissue tension. Furthermore, wearing tight jeans constrains pelvic and hip movement. Yoo and Yoo [23] observed significant limitations in hip movement caused by tight jeans, resulting in increased lumbar spine movement to complete trunk flexion, potentially altering FRP patterns [19]. Chen et al. [24] highlighted how jeans restrict pelvic activity, affecting bending and lifting mechanics. Previous findings suggest that pelvic movement affects the occurrence of the FRP. However, a systematic evaluation of how leg posture impacts FRP remains lacking. When standing with knees fully extended, leg muscles, including the lower leg flexor muscles, contribute to maintaining ankle torque [25]. These muscles play a crucial role in maintaining human body balance, thereby influencing FRP during deep trunk flexion. Therefore, in addition to the lumbar erector spinae (LES), several specific muscle groups in the lower limbs related to leg movements are typically examined based on the research purpose. These include the biceps femoris (BF) [4,17,26,27,28,29] and the gastrocnemius (GAS) [4,30,31]. Other muscles associated with the pelvis and lower limbs were also evaluated to determine their roles during trunk flexion. However, the BF and GAS are generally considered the primary muscles for assessing hip extension, knee flexion, and knee stabilization [32].

The relationship between lower limb posture and the lumbar spine can evidently impact back muscle FRP. Some studies limit lower limb or pelvic movement during FRP assessment [16,17,19,22,33,34,35], while others solely control trunk flexion [15,21,36,37,38]. The critical disparity lies in the constrained leg condition, precisely controlling trunk position (including lower limb and pelvic influences), whereas the unconstrained condition evaluates FRP during trunk forward bending in a natural leg posture. These variations in experimental control variables may impact results, complicating comparisons between studies when examining FRP differences.

This study aimed to elucidate how different lower limb control conditions affect back muscle FRP, potentially leading to varied FRP patterns. It was hypothesized that limiting lower limb movement would prompt distinct FRP patterns in back muscles due to postural balance requirements and increased lower limb muscle activity, given the interconnectedness of the lower limbs, pelvis, and lumbar spine. These findings would offer valuable insights for practical applications in research contexts.

## 2. Materials and Methods

To explore the influence of lower limb postural control on back muscle FRP, we enlisted 32 male participants and conducted an FRP experiment. Trunk movements were recorded from upright (0°) to forward bending at 15° intervals up to 90°, under two leg postures (constrained and unconstrained). Muscle activities of the LES, BF, and GAS, as well as the lumbosacral angle (LSA), were measured. Notably, the long and medial heads of the BF and GAS muscles were specifically chosen for the test. All procedures adhered to the 2013 World Medical Association Declaration of Helsinki, and informed consent was obtained from all participants.

### 2.1. Participants

This study involved 32 male university students, aged 19–24 years, all with right-hand and right-leg dominance, and no history of musculoskeletal injuries or back/leg pain. Participants’ dominance was verified by referring to previous studies [39,40]. All participants received compensation of approximately USD 50 for their participation. They were instructed to avoid strenuous activities and late nights during the experimental period to ensure their body was in normal condition during the experiment. Prior to commencing the experiment, participants were required to self-report and confirm their adherence to these guidelines. Participants were categorized into low- and high-flexibility groups (16 participants each) using the toe-touch flexibility test, adapted from Shin et al. [4] and Ayala et al. [41]. In this test, participants bent forward from a standing position and attempted to touch the ground with their fingertips (Figure 1). Those who reached 3 cm or more below the floor baseline were placed in the high-flexibility group, while those who did not reach the floor baseline by 3 cm or more were placed in the low-flexibility group. Initially, 48 candidates were screened, as illustrated in Figure 2. After the toe-touch test, 17 individuals met the low-flexibility criteria, and 19 met the high-flexibility criteria. To ensure balanced participant characteristics between the two test groups, a total of 32 individuals (16 in each flexibility level) ultimately participated in the FRP experiment. This approach was primarily intended to avoid potential interference from those with middle flexibility on the research results.

Table 1 presents the anthropometric data for each group. An independent *t*-test showed no significant differences between the groups in any variable (*p* > 0.05) except flexibility. The mean (standard deviation) flexibility values were −12.1 cm (8.3 cm) for the low-flexibility group and 9.2 cm (5.9 cm) for the high-flexibility group, with an average difference of 21.3 cm between the groups. Anthropometric data, particularly heights such as acromial, hip, and knee height, which were indistinguishable between the two flexibility groups, helped prevent interference in the test data caused by differences in body size.

### 2.2. Electromyography

The TeleMyo 2400, an electromyography (EMG) device from Noraxon (Scottsdale, AZ, USA), was used to measure the activation of the LES, BF, and GAS muscles on each participant’s dominant side. The procedures for EMG testing, which included skin preparation, electrode placement, and fixation, as well as data acquisition and processing, adhered to the Surface Electromyography for the Non-Invasive Assessment of Muscles (SENIAM) guidelines [42]. Ag/AgCl surface electrodes, with a 10 × 10 mm^2^ lead-off area and a center-to-center distance of approximately 20 mm, were placed parallel to the muscles. Before electrode application, skin impedance was minimized by shaving excess body hair (if necessary), gently abrading the skin with fine-grade sandpaper, and wiping the skin with alcohol swabs, following SENIAM guidelines. According to SENIAM protocols, the placements of electrodes for the investigated muscles in this study were as follows: (1) LES muscle: the electrodes were positioned 2 finger widths lateral to the spinal process of L1; (2) BF muscle: the electrodes were placed at 50% on the line between the ischial tuberosity and the lateral epicondyle of the tibia; and (3) GAS muscle: the electrodes were placed on the most prominent bulge, aligned with the direction of the leg. In the test, the reference electrode was placed around the ankle.

Prior to EMG data collection, participants engaged in standardized muscle-specific maximum voluntary contraction (MVC) exercises to normalize the EMG data for each trial. The MVC testing protocols were based on Vera-Garcia et al. [43]. For the LES muscles, participants lay prone on a bench with their torsos supported and legs hanging off, exerting maximal effort to extend their lower trunk and hips against manual resistance. For the BF muscles, participants lay prone with knees flexed at 30°, applying maximum effort against manual resistance. For the GAS muscles, participants performed an isometric contraction against resistance in the direction of ankle plantar flexion. Strong verbal encouragement was provided throughout each MVC measurement. Each participant performed three MVC trials per muscle, maintaining each contraction for at least 5 s, with a 3 min rest period between trials. The highest EMG amplitude for each muscle, calculated using a 0.5 s moving average window [43], was used as the MVC value for subsequent analysis [44].

Afterward, the electrical signals from both the MVC tests and experimental trials were filtered using an analog band-pass filter set between 20 and 600 Hz, and then sampled at a rate of 1200 Hz [42]. To obtain integrated EMG (IEMG) data, the sampled signals were fully rectified and processed. A normalization process was then performed to compare IEMG data from the experimental trials with MVC IEMG data over an identical 5 s interval. All muscle activation values were expressed as percentages of the MVC IEMG data (i.e., %MVC).

### 2.3. Lumbar Spine Curvature Measurements

Participants’ spinal curvature was evaluated by measuring the LSA as they stood upright and flexed their trunk in 15° increments from 0° to 90°. The trunk angle was calculated using the line from the acromial shelf to the hip relative to the vertical axis. Before data collection, four adhesive reflective markers were affixed to specific body joints (shoulder, hip, knee, and ankle), along with two stick markers on the skin over the first lumbar and first sacral spinous processes (see Figure 3). The external LSA (ELSA), recorded for each trial using stick markers at S1 and L1, was utilized to estimate the internal LSA using prediction models developed by Chen and Lee [45] and employed in prior studies by Chen et al. [16,17]. Each participant’s ELSA at a specific trunk position was subsequently used as a reference to calculate the internal LSA. The models are expressed as follows:IL_1_ = 0.988 × SL_1_ + 3.627 (*R*^2^ = 0.968)
IS_1_ = 0.734 × SS_1_ + 29.678 (*R*^2^ = 0.916)
where SL_1_ and SS_1_ represent the respective angles of the external stick markers L_1_ and S_1_, and IL_1_ and IS_1_ represent the internal angles. The internal LSA can be obtained by determining the angle between IL_1_ and IS_1_ [45].

### 2.4. Experimental Design and Procedure

This study evaluated the muscle activities in participants’ lower back and legs, as well as the LSA, during trunk flexion at various angles. Prior to the experiment, participants were briefed on and familiarized with the procedures. Each participant performed the tests under both constrained and unconstrained leg postures, as illustrated in Figure 3. Trunk flexion angles ranged from 0° to 90° in 15° increments. Throughout the experiment, participants were instructed to flex their trunk from an upright position to six specific angles while maintaining straight knees and keeping their hands crossed on their chests. Each participant repeated each test combination twice for reliability, and the mean values were used for further analysis. Static EMG and LSA data were collected for each participant across 28 test combinations (2 leg postures × 7 trunk flexion positions × 2 repetitions). The order of these 28 combinations was randomized for each participant to minimize experimental error.

During the test, a MacReflex motion analysis system (Qualisys, Göteborg, Sweden) was set up approximately 5 m to the right-lateral side of the participant and perpendicular to their sagittal plane to capture the 2D marker positions (resolution = 1:30,000 in the camera field of view at 120 Hz, with signals being low-pass filtered at 6 Hz). In this study, when the leg posture was restricted, participants needed to maintain an upright leg posture, with the hip and ankle joints aligned vertically. When the leg posture was not restricted, participants adopted a free leg posture. The postural difference between the two leg conditions is visually illustrated as PD in Figure 3. During the static-posture test, participants were instructed to flex their trunk naturally, following the method used by Chen et al. [16]. To ensure accuracy, the experimenter confirmed that the participant’s trunk line (connecting the shoulder and hip markers) and leg line (connecting the hip and ankle markers, when in a constrained leg condition) matched the preset lines on the feedback monitor of the motion analysis system. Participants were verbally guided by the experimenter to flex their trunk until they achieved the desired position with either a constrained or unconstrained leg posture. During the test, participants were instructed to bend their trunks from an upright position to the specified angles as slowly as possible to minimize the impact of movement speed [15].

Upon assuming the required trunk flexion and leg posture, a trigger signal initiated the simultaneous collection of motion and EMG data to ensure synchronization. Participants maintained each posture for at least 5 s [19], with data collected for the full 5 s duration of each position for analysis. A minimum rest period of 3 min was enforced between successive trials to minimize potential muscle fatigue and passive tissue creep, and no participant underwent testing for more than 1.5 h in total.

### 2.5. Statistical Analysis

Statistical analyses were performed using SPSS version 22.0 (SPSS, Inc., Chicago, IL, USA), with a significance level set at α = 0.05. Normal distribution of numerical variables was assessed using the Kolmogorov–Smirnov test, while homogeneity of variances was evaluated using Levene’s test to ensure robustness of the analysis. A three-way repeated-measures analysis of variance (ANOVA) was conducted to explore the effects of individual flexibility, leg posture, and trunk flexion on the dependent variables (muscle activations and LSA). Each participant was treated as a block and underwent all treatment combinations in a randomized order. Flexibility was considered a between-subject factor, while trunk angle and leg posture variables served as within-subject factors. Post hoc comparisons were conducted using the Duncan multiple-range test (MRT). Additionally, the independent *t*-test was employed to assess statistically significant differences in muscle activations and LSA between the two flexibility groups or between the two leg postures for each trunk flexion position.

## 3. Results

The results of the three-way ANOVA indicated a significant impact of independent variables on all measured responses, as outlined in Table 2. When averaged across other variables (i.e., leg posture and trunk flexion angle), the main effects revealed that greater flexibility was associated with heightened muscle activities of the LES (8.0 %MVC) and BF (6.6 %MVC), but decreased GAS activity (6.1 %MVC), and increased LSA (13.6°), compared to lower flexibility (all *p <* 0.01; 7.3%, 5.9%, 7.2 %MVC, and 11.3°, respectively). Similarly, when participants adopted a constrained leg posture, LES activities (7.1 %MVC vs. 8.3 %MVC; *p <* 0.001) and LSA (11.0° vs. 14.0°; *p <* 0.001) were significantly lower compared to when using an unconstrained leg posture. Conversely, higher BF (6.6 %MVC vs. 5.9 %MVC; *p <* 0.01) and GAS activities (8.0 %MVC vs. 5.3 %MVC; *p <* 0.001) were observed in leg constrained conditions.

As presented in Table 2, the interactions of leg posture × trunk angle significantly affected both LES (*p <* 0.05) and GAS activities (*p <* 0.01). Figure 4 and Figure 5 illustrate the cross-analyses for the LES and leg muscle activities under different test combinations, respectively. In the unconstrained leg position, the LES activity was notably higher when the trunk flexed forward at 60° compared to when the legs were constrained. This difference was statistically significant for both the high-flexibility group (*p <* 0.01) and the low-flexibility group (*p <* 0.05). Except for standing with the trunk upright, the GAS activity was notably lower when the legs were unconstrained compared to when they were constrained. Conversely, BF activity exhibited a slight increase in the constrained leg posture during trunk flexion.

Table 3 and Table 4 demonstrate the LSA changes (Duncan MRT) under various trunk angles for different flexibility and leg posture groups, respectively, when averaged across the other variable. The effect of the flexibility variable on LSA was statistically significant when the trunk flexed at 30° and 45° (*p <* 0.05), while the effect of leg posture on LSA was significant during trunk flexion at 45° (*p <* 0.01), 60° (*p <* 0.01), and 75° (*p <* 0.05). Figure 6 visually presents the comparative analysis of LSAs across different test combinations at varying trunk angles.

## 4. Discussion

This study delved into the effects of the two leg postures (natural or constrained) on back muscle FRP, a topic not previously explored. From an ergonomic and practical standpoint, investigating various trunk forward bending angles when FRP occurs holds relevance for on-site operations or task design. It highlights how the interconnected movements between the trunk and lower limbs influence FRP, particularly its occurrence and magnitude. This diverges from the clinical application of FRP as an indicator for diagnosing patients’ LBP, which typically focuses on maximum or near-maximum trunk forward flexion [20,21,46,47]. The primary contribution of this study lies in clarifying that when leg posture is constrained, as expected, it may lead to an overestimation of the degree of FRP in the back muscles compared to the natural leg posture. The study findings provide reference points for subsequent research on FRP-related topics.

The ANOVA results revealed that all independent variables had a significant main effect on the measured responses (Table 2). Constraining the leg posture resulted in a more pronounced FRP of the LES muscle. Figure 4 illustrates the LES activities for four combinations of flexibility and leg posture at different trunk flexion angles. It is evident from the figure that when the trunk was bent forward at 45°, constraining the leg posture reduced LES activity, irrespective of whether it was in the high-flexibility (*p <* 0.01) or low-flexibility group (*p <* 0.05). Notably, when the high-flexibility group adopted a natural leg posture, FRP might even be delayed until the trunk flexed at 60°, a significant difference compared to the low-flexibility group (*p <* 0.05). Previous studies have indicated that individuals with low flexibility exhibit earlier onset of FRP [16,17,37] and lower BF muscle activity [24] than those with high flexibility, a phenomenon observed in this study as well. Table 2 also indicates that the interaction of leg posture and trunk angle significantly influenced LES activity (*p <* 0.05). As depicted in Figure 4, when trunk forward flexion is less than 45°, LES activities remain relatively constant (when averaged across data from high- and low-flexibility groups). However, within trunk forward flexion angles of 45° to 75°, there is a significant difference in LES activities between the two leg postures, particularly at 60°, where constrained and unconstrained LES activities are 7.1 %MVC and 10.5 %MVC, respectively, with a difference of 3.4 %MVC. Considering the flexibility variable, this difference increases to 5.3 %MVC.

Nordin et al. [48] discovered that when the knees are straight and the trunk is flexed forward, the initial 50–60° of trunk flexion primarily involves lumbar spine movement, followed by forward pelvic rotation to complete the flexion action. Solomonow et al. [8], through a combination of movement and EMG measurements, determined that the onset of FRP in the back muscles of healthy subjects typically occurs around 45–50° of trunk forward bending, consistent with the findings of this study. However, some studies have reported earlier occurrences of FRP [49]. Figure 6 illustrates the change in LSAs at different trunk flexion angles. The collective effect of leg posture and individual flexibility on LSA remains relatively consistent, wherein larger LSAs correspond to higher LES activities, suggesting a relatively lower degree of FRP. In real-world material handling operations, previous studies have noted that trunk forward flexion angles typically fall within the range of 30° to 60° [50,51]. Therefore, assessing FRP characteristics within this trunk flexion range holds particular significance. This study revealed that the influence of leg posture on FRP predominantly manifests within this range, highlighting the crucial connection between the leg posture variable and actual working postures.

This study is subject to several limitations. Firstly, despite the participation of 32 young individuals in the experiment, the grouping resulted in only 16 individuals in each group. The relatively small sample size represents a primary limitation of this study, and its findings may not be broadly applicable, particularly to the female population, considering potential gender differences in flexibility [52]. Future studies could enhance their validity by increasing the number of participants and including individuals with diverse demographic characteristics for more comprehensive generalization. Additionally, this study did not include males with middle flexibility (scores between ±3 cm, as shown in Figure 1 and Figure 2); instead, it focused on recruiting two extreme flexibility groups. Shin et al. [4] employed a mid-flexibility group in addition to low- and high-flexibility groups, noting that LES FRP in mid-flexibility participants fell proportionally between the two extremes, especially at a trunk flexion angle of 90°. Furthermore, this study revealed that FRP is based on the specific and non-continuous trunk angles examined. In our study, we measured the activities of the LES, BF, and GAS muscles. Instead of focusing on muscle activity, this study attempted to use changes in lumbar lordosis to evaluate how pelvic movement influences FRP. Further investigation is needed to understand the impact of other muscle groups in the lower limbs, especially those involved in pelvic movement, on FRP-related tests.

## 5. Conclusions

This study examined the activities of lower back and leg muscle groups, along with changes in lumbar lordosis, across different trunk angles and leg postures. The findings revealed that when trunk flexion was not considered in conjunction with leg posture, the degree of FRP in back muscles decreased, and FRP onset could be delayed in individuals with high flexibility. Conversely, when leg posture was constrained, back muscle FRP tended to be overestimated, with LES activities closely associated with the measured LSAs. These results suggest that when applying back muscle FRP to on-site manual work evaluations, caution is needed, as laboratory protocols that control leg posture may lead to an overestimation of the FRP degree. Therefore, careful interpretation of the results is warranted.

## Figures and Tables

**Figure 1 bioengineering-11-00736-f001:**
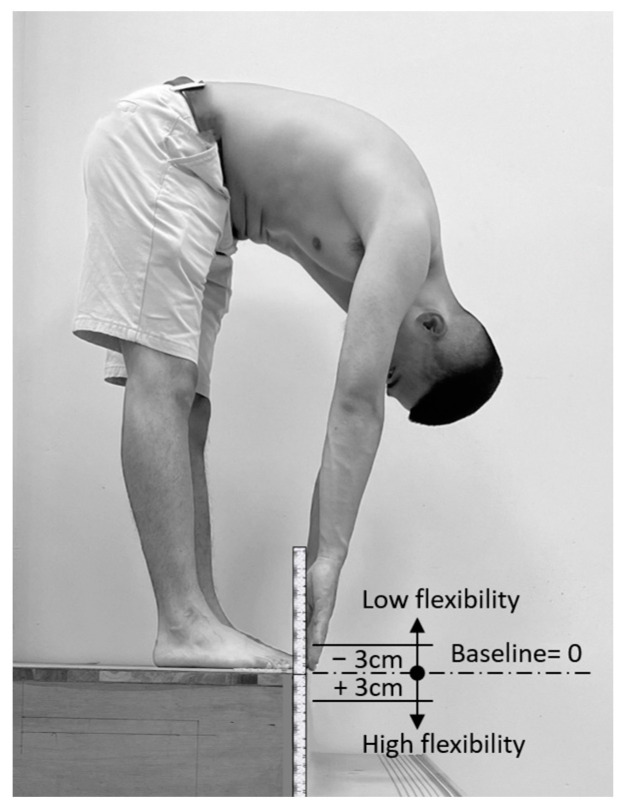
Schematic of criteria for determining individual flexibility.

**Figure 2 bioengineering-11-00736-f002:**
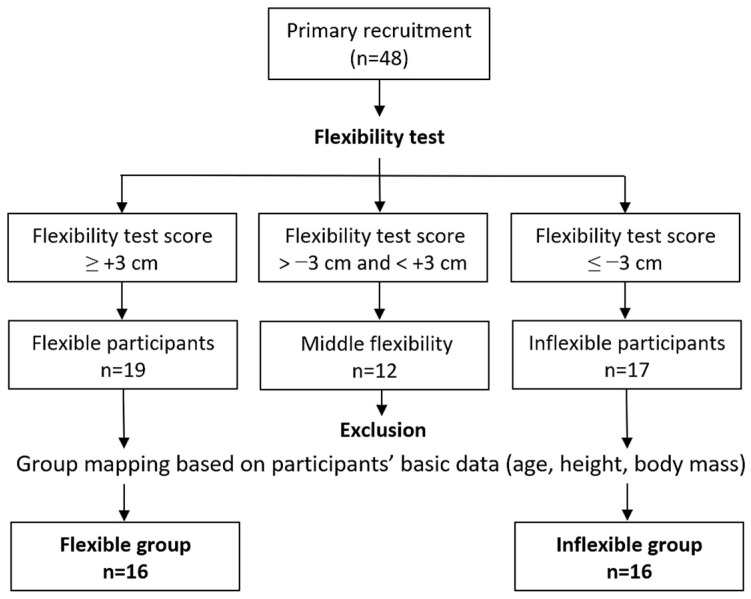
Flowchart depicting the selection process for the two test groups enrolled in the study.

**Figure 3 bioengineering-11-00736-f003:**
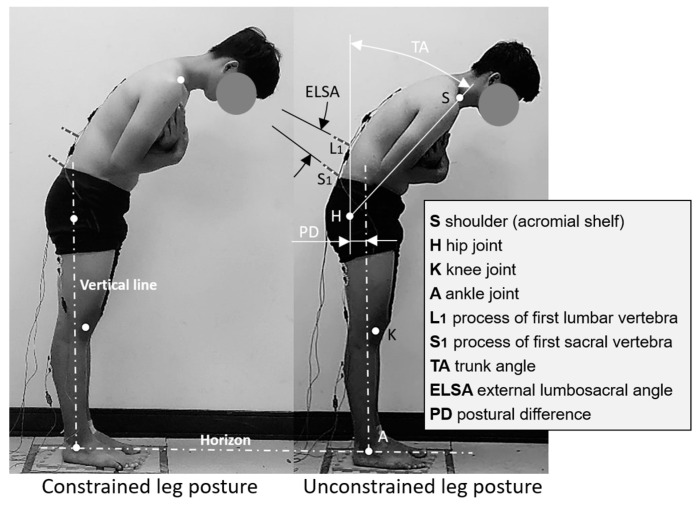
Schematic illustration demonstrating the testing posture, body angles during trunk flexion, and positions of markers and stickers on the participant’s body.

**Figure 4 bioengineering-11-00736-f004:**
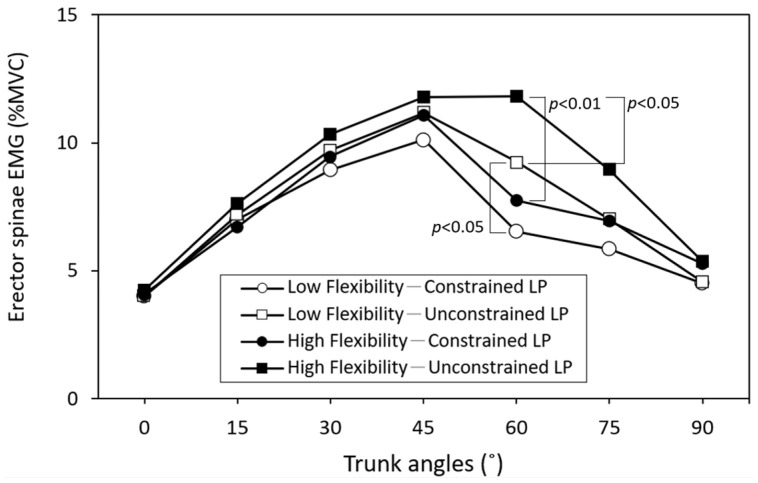
Lumbar erector spinae (LES) activities across different trunk flexion angles, with comparisons using independent *t*-tests between two leg postures (LP) for each flexibility group.

**Figure 5 bioengineering-11-00736-f005:**
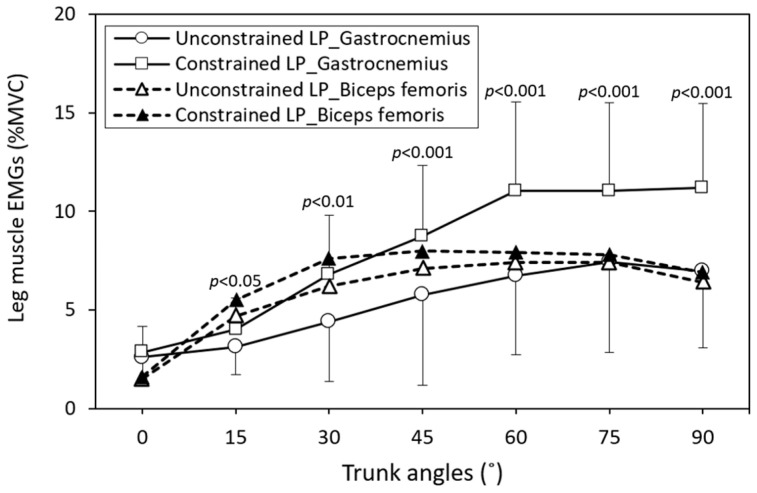
Leg muscle activities across various trunk flexion angles between the two leg postures (LP), with comparisons using independent *t*-tests for gastrocnemius electromyography.

**Figure 6 bioengineering-11-00736-f006:**
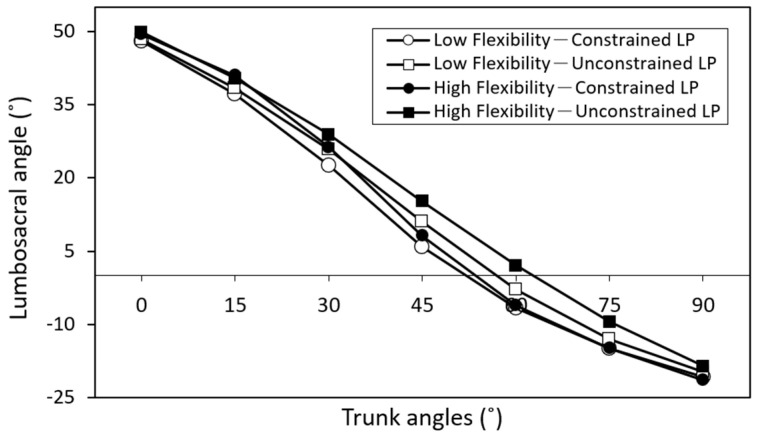
Comparisons of lumbosacral angles (LSAs) for four test combinations comprising two flexibility levels and two leg postures (LP).

**Table 1 bioengineering-11-00736-t001:** Fundamental data of the two test participant groups distinguished by flexibility levels.

Items	Low Flexibility (n = 16)		High Flexibility (n = 16)	
	Mean (SD)	Range	Mean (SD)	Range
Age (years)	21.6 (1.4)	19–24	21.6 (1.5)	20–24
Height (cm)	172.8 (4.7)	166–180	173 (4.5)	161–178
Body mass (kg)	67 (7.8)	52–77	68.1 (7.5)	49–78
Acromial height (cm)	142 (4.7)	135–150	141.9 (4.2)	135–151
Hip height (cm)	88.4 (4.6)	78–95	87.0 (3.6)	79–93
Knee height (cm)	47.8 (2.4)	44–52	47.5 (1.8)	44–50
Flexibility (cm)	−12.1 (8.3)	−3–−34	9.2 (5.9)	3–25

Note: Data were presented in mean (standard deviation, SD).

**Table 2 bioengineering-11-00736-t002:** Results of the three-way analysis of variance (ANOVA) indicating the impact of independent variables on related muscle activations and lumbosacral angle.

Variables	Responses	DF	SS	MS	F	*p*	Power
Flexibility	Lumbar erector spinae	1	66	66	7.2	<0.01	0.763
	Biceps femoris	1	45	45	8.6	<0.01	0.835
	Gastrocnemius	1	119	119	9.5	<0.01	0.869
	Lumbosacral angle	1	506	506	9.7	<0.01	0.876
Leg posture	Lumbar erector spinae	1	107	107	11.6	<0.001	0.926
	Biceps femoris	1	45	45	8.5	<0.01	0.830
	Gastrocnemius	1	759	759	60.7	<0.001	1.000
	Lumbosacral angle	1	901	901	17.3	<0.001	0.986
Trunk angle	Lumbar erector spinae	6	2007	335	36.4	<0.001	1.000
	Biceps femoris	6	1646	274	52.6	<0.001	1.000
	Gastrocnemius	6	2625	437	35.0	<0.001	1.000
	Lumbosacral angle	6	234,761	39,127	753.3	<0.001	1.000
Flexibility × Leg posture	Lumbar erector spinae	1	4	4	0.4	0.505	0.102
	Biceps femoris	1	<1	<1	<0.1	0.995	0.050
	Gastrocnemius	1	2	2	0.2	0.687	0.069
	Lumbosacral angle	1	41	41	0.8	0.374	0.144
Flexibility × Trunk angle	Lumbar erector spinae	6	37	6	0.7	0.668	0.270
	Biceps femoris	6	43	7	1.4	0.222	0.539
	Gastrocnemius	6	19	3	0.3	0.956	0.120
	Lumbosacral angle	6	105	18	0.3	0.917	0.146
Leg posture × Trunk angle	Lumbar erector spinae	6	116	19	2.1	<0.05	0.757
	Biceps femoris	6	17	3	0.6	0.770	0.221
	Gastrocnemius	6	226	38	3.0	<0.01	0.907
	Lumbosacral angle	6	479	80	1.5	0.165	0.593
Flexibility × Leg posture × Trunk angle	Lumbar erector spinae	6	7	1	0.1	0.992	0.083
	Biceps femoris	6	3	1	0.1	0.996	0.076
	Gastrocnemius	6	9	2	0.1	0.994	0.080
	Lumbosacral angle	6	104	17	0.3	0.920	0.144

**Table 3 bioengineering-11-00736-t003:** Lumbosacral angle categorized into Duncan groups across two flexibility levels (unit: °, n = 32).

Trunk Angle (°)	Low Flexibility	Duncan Groups	High Flexibility	Duncan Groups	Difference
0	48.1 (5.8)	A	49.6 (8.3)	A	1.5
15	37.8 (6.5)	B	40.6 (7.3)	B	2.8
30	24.2 (6.8)	C	27.6 (7.0)	C	3.4 *
45	8.5 (6.6)	D	11.7 (6.5)	D	3.2 *
60	−4.8 (7.0)	E	−2.0 (8.7)	E	2.8
75	−14.0 (6.1)	F	−12.2 (7.6)	F	1.8
90	−20.3 (7.1)	G	−20.0 (7.4)	G	0.3

Note: Data in mean (standard deviation) with the same letter do not differ in Duncan’s test; * *p* < 0.05.

**Table 4 bioengineering-11-00736-t004:** Lumbosacral angle categorized into Duncan groups across different leg postures (unit: °, n = 32).

Trunk Angle (°)	Constrained Leg Posture	Duncan Groups	Unconstrained Leg Posture	Duncan Groups	Difference
0	48.7 (7.4)	A	49.0 (6.9)	A	0.3
15	39.1 (7.9)	B	39.4 (6.1)	B	0.3
30	24.4 (7.3)	C	27.3 (6.8)	C	2.9
45	7.1 (8.0)	D	13.1 (7.6)	D	6.0 **
60	−6.4 (7.4)	E	−0.4 (7.4)	E	6.0 **
75	−14.9 (6.3)	F	−11.1 (7.1)	F	3.8 *
90	−21.1 (6.8)	G	−19.2 (7.6)	G	1.9

Note: Data in mean (standard deviation) with the same letter do not differ in Duncan’s test; * *p* < 0.05, ** *p* < 0.01.

## Data Availability

The data are available upon reasonable request to the corresponding author.

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
