# Peer review of "Differential Back Muscle Flexion–Relaxation Phenomenon in Constrained versus Unconstrained Leg Postures"

_bioengineering, 2024, doi:10.3390/bioengineering11070736_

Round 1
Reviewer 1 Report
Comments and Suggestions for Authors
Thank you for the opportunity to review the manuscript titled "Differential back muscle flexion-relaxation phenomenon in constrained versus unconstrained leg postures” Thea im of the study was to explore the influence of leg posture control and individual flexibility on back muscle e flexion-relaxation phenomenon (FRP). A total of 32 men were examined. Based on the obtained research results, the authors conclude that using an inappropriate leg posture during back muscle FRP assessments can overestimate FRP. These findings offer guidance for designing future FRP research protocols.
Before this manuscript reaches the level of being considered for publication, many elements must be corrected and clarified. At this stage, many places are unclear and difficult to understand.
Line 12. The authors write: „This study aimed to Explorer….flexibility on back muscle FRP.” Were only the spinal extensors tested? This misleads the reader!
Line 54-57. What does this have to do with the topic of the work?
Line 70-74. This is the methodical part of the work, not the introductory part!
Line 85: The authors write: „…avoid strenuous activities and late nights…” Was it controlled and checked in any way and how?
Line 91-93: The authors write: „Those who reached 3 cm or more below the floor baseline were placed in the high-flexibility group, while those who did not reach the floor baseline by 3 cm or more were placed in the low-flexibility group.” What about those who were between 3 cm to the ground and 3 cm below the ground? They were excluded from the study. Does this description show that or is it unclear? If they were excluded, some flow diagram would be useful. Weren't there people in this group between these results? This is unlikely!
Table 1. What are the parameters: Acromial height, Hip height, Knee height intended for?
Line 106. How was the dominant side checked? Just because they were right-handed doesn't mean they were right-footed! There is no description or I didn't notice it.
Line 113-116: For HAM measurement at an electrode on the LS spine? For LES measurement on the back of the thigh? Is this a correct description of electrode application?
Line 116: Between the lateral and medial heads of GAS? Or where?
Line 139: What measurement tool was used to measure LSA? For me this description is not very clear! How was it controlled when the subject should stop while bending? What tool?
Line 308: The authors write: „Additionally, this study did not include males with normal flexibility;” These people were rejected from the group? No flow diagram! It is unlikely that there would be no people with "normal" flexibility in such a group!
Line 316-328: The conclusions are too descriptive and too long.
Reviewer 2 Report
Comments and Suggestions for Authors
This study has presented the influence of leg posture control and individual flexibility on back muscle FRP. The authors recruited 32 male participants and analysed activities of the erector spinae, hamstrings, and gastrocnemius muscles alongside the lumbosacral angle (LSA) when the trunk flexion from 0° to 90° in 15° increments under two leg postures, enabling an analysis of FRP and its correlation with the investigated variables. The findings highlight the significant effects of all examined factors on the measured responses. In short, this study's findings provide reference points for subsequent research on FRP-related topics. Some suggestions are as follows:
1. In section 2.1, the measurement process of the high and low flexibility groups will be easier to comprehend with the help of a figure.
2. There seems to be a contradiction between Figure 1 and the description in section 2.4, which states that participants placed their hands over the navel area. This paradox needs to be addressed to avoid confusion and ensure that readers understand the research process clearly.
3. Figure 1 does not clearly depict the two types of leg postures. A more detailed description of these postures in the 2.1 participants section will provide a better understanding of the research setup, thereby instilling a sense of reassurance and confidence in the accuracy of the research among our readers.
4. It is better to add the sampling frequency of the MacReflex motion analysis system (Qualisys, Göteborg, Sweden) in section 2.4.
Reviewer 3 Report
Comments and Suggestions for Authors
The main purpose of the author is to explore the changes of lower limb muscles under different movements. However, there are still several questions that need to be further solved by the author.
First, why did the author choose these muscles as the target to explore, and why did not choose the soleus, gluteus maximus and other posterior chain muscles? Please give a reasonable explanation in the introduction.
Secondly, please add the complete processing process of EMG in the method section. And give a clear anatomical location for placing EMG sensors.
Finally, and this is the part that I am most confused about, the hamstrings are a muscle group on the back of the thigh, but when we usually collect muscle data, we will place it in the middle of each muscle to collect data. So, how did the author collect this large muscle group? Please give a reasonable explanation.
Round 2
Reviewer 1 Report
Comments and Suggestions for Authors
After re-reading the manuscript and the authors' responses to the review, I believe that it now meets the conditions for publication in the journal Bioengineering
Reviewer 3 Report
Comments and Suggestions for Authors
none